# Berberine in Bowel Health: Anti-Inflammatory and Gut Microbiota Modulatory Effects

**DOI:** 10.3390/ijms262412021

**Published:** 2025-12-13

**Authors:** Anna Duda-Madej, Szymon Viscardi, Jakub Piotr Łabaz, Ewa Topola, Wiktoria Szewczyk, Przemysław Gagat

**Affiliations:** 1Department of Microbiology, Faculty of Medicine, Wroclaw Medical University, Chałubińskiego 4, 50-368 Wroclaw, Poland; 2Faculty of Medicine, Wroclaw Medical University, Ludwika Pasteura 1, 50-367 Wroclaw, Poland; szymon.viscardi@student.umw.edu.pl (S.V.); jakub.labaz@student.umw.edu.pl (J.P.Ł.); ewa.topola@student.umw.edu.pl (E.T.); wiktoria.szewczyk@student.umw.edu.pl (W.S.); 3Faculty of Biotechnology, University of Wroclaw, Fryderyka Joliot-Curie 14a, 50-137 Wroclaw, Poland

**Keywords:** anti-inflammatory, berberine, colitis, gut microbiota, invasive pathogens

## Abstract

Disruption of the gut-microbiome-brain axis contributes to the development of chronic inflammation, impaired intestinal barrier integrity, and progressive tissue damage, ultimately reducing quality of life and increasing risk of comorbidities, including neurodegenerative diseases. Current therapies are often limited by adverse effects and insufficient long-term efficacy, highlighting the need for more comprehensive therapeutic approaches. Berberine (BRB), a plant-derived isoquinoline alkaloid, has attracted growing attention due to its pleiotropic immunomodulatory, neuroprotective, and gut-homeostasis-modulating properties, which involve reshaping the gut microbiota and underscore its therapeutic relevance within the gut–microbiome–brain axis. The aim of this review is to synthesize current scientific evidence regarding the anti-inflammatory mechanisms of BRB in inflammatory bowel disease (IBD). We compare its activity with first-line therapies and discuss its impact on microbial composition, including the bidirectional regulation of specific bacterial taxa relevant to intestinal and systemic disorders that originate in the gut. Furthermore, we emphasize that gut bacteria convert BRB into bioactive metabolites, contributing to its enhanced intraluminal activity despite its low systemic bioavailability. By integrating molecular and microbiological evidence, this review fills a critical knowledge gap regarding the comprehensive therapeutic potential of BRB as a promising candidate for future IBD interventions. The novelty of this work lies in unifying fragmented findings into a framework that explains how BRB acts simultaneously at the levels of host immunity, microbial ecology, and neuroimmune communication—thus offering a new conceptual model for its role within the gut–microbiome–brain axis.

## 1. Introduction

The gut is not coincidentally referred to as the “second brain.” It communicates with our brain through three different pathways: (i) neuronal, via neurons; (ii) hormonal, through cortisol, adrenaline, and serotonin; and (iii) immune, via cytokines [1,2]. The resulting gut–brain axis is a bidirectional pathway: (i) from top to bottom—secretion of signaling molecules modulating gut physiology, and (ii) from bottom to top—production of neurotransmitters or short-chain fatty acids sending signals through the gut [3]. Therefore, when the gut function is impaired, the entire organism suffers. Intestinal diseases are among the most common diseases of the gastrointestinal tract [4]. They can manifest as: (i) inflammatory bowel disease (IBD), e.g., ulcerative colitis—UC, Crohn’s disease—CD, indeterminate colitis—IC [5]; (ii) infectious colitis, caused by viruses or bacteria [6]; (iii) microscopic colitis, associated with gut microbiota, genetic predispositions, and bile acid and fatty acid malabsorption disorders [7]; and (iv) ischemic colitis, which is insufficient blood flow to the intestinal wall [8]. These conditions inevitably lead to a deterioration in quality of life, disability, and even premature death [9,10].

The pathogenesis of IBD involves superficial inflammation in UC, originating from these “ulcers,” while Crohn’s disease progresses throughout the entire length of the intestine and is accompanied by the presence of, among other things, granulomas. Numerous scientific studies conducted in recent years indicate that IBD results not only from genetic predispositions, prenatal and childhood life exposures, and environmental, dietary, and lifestyle factors but, most importantly, from complex interactions between the human body and gut microbiota [11,12]. The underlying process is unclear, but it is likely related to factors that result in an abnormal immune response and, consequently, intestinal dysbiosis [13]. This unique connection forms a new network, known as the gut–microbiome–brain axis (GMBA) [14,15,16]. Within this relationship, disruptions in microbiota function can impair the integrity of the gut barrier, the functioning of the immune system, preventing the growth of harmful microorganisms, and overall maintaining the body’s homeostasis [17,18]. These irregularities lead to inflammation, which is the causative factor in most, if not all, of the above-mentioned conditions. Its role is to protect the body and restore balance by eliminating the causative agents and repairing tissue damage [19,20]. However, when the harmful factor originates from the microbiota, the inflammatory process develops slowly and excessively, transitioning into a chronic state. This chronic inflammation is characterized by uncontrolled overproduction of pro-inflammatory cytokines and lipids, as well as activation of signaling pathways that can finally lead to irreversible tissue damage and chronic gastrointestinal diseases [21,22].

Currently, many anti-inflammatory drugs are known to be used for IBD. However, the side effects of their application and efficacy are under constant controversy. The anti-inflammatory therapy is largely based on the use of immunosuppressive, immunomodulatory, and anti-inflammatory drugs. Among the oldest maintenance therapy drugs are sulfonamide derivatives (e.g., sulfasalazine, mesalazine), sulfones (e.g., dapsone), and salicylate derivatives (e.g., 5-aminosalicylic acid) [23]. Typical immunosuppressive drugs include cytostatic antimetabolites, such as azathioprine, methotrexate, and steroidal anti-inflammatory drugs (e.g., prednisolone, methylprednisolone). Importantly, the use of these drugs is limited by serious side effects, such as methemoglobinemia, blood disorders (including pancytopenia), hepatotoxicity, immunosuppression, myelotoxicity, and the risk of acute pancreatitis [24,25]. Therefore, the development of biological drugs, particularly those from the monoclonal antibody group, is especially promising at present. Some of them, i.e., infliximab, adalimumab, etanercept (anti-TNF-α), risankizumab (anti-IL-23), have already been registered for the treatment of IBD [26]. Additionally, recent reports on the efficacy of upadacitinib (a Janus kinase inhibitor) in treating Crohn’s disease at the signaling cascade level open up new possibilities for the therapy of IBD [27].

Overall, these efforts represent only a small contribution toward addressing the growing need for treatments for these globally concerning diseases, the prevalence of which is increasing every year [28]. Furthermore, recent studies indicate that IBD, due to the bacterial dysbiosis occurring during its progression, contributes to the development of neurodegenerative diseases such as Alzheimer’s (AD) and Parkinson’s disease (PD) [16]. These conditions, in turn, pose a significant societal threat as their incidence is rising rapidly. Therefore, the search for new drugs that relieve the symptoms of acute inflammation, thereby protecting against the emergence of a chronic condition, remains of the utmost importance.

Nature offers a wide array of diverse compounds, with particular promise shown by plant-derived isoquinoline alkaloid berberine (BRB; Figure 1), naturally present in plants from the *Berberidaceae* family [29]. The compound is widely known for its anti-inflammatory [30], anti-cancer [31], bactericidal [32], and antifungal [33] properties. Its widespread availability and broad range of activities make BRB very interesting from the medical point of view, particularly its potential to benefit the nervous system, in addition to its various anti-inflammatory effects. Recent studies also suggest that BRB is capable of delaying processes leading to neurodegenerative diseases [34,35,36]. Such promising activities make BRB a potential candidate for a new anti-inflammatory and immunomodulatory drug for the treatment of IBD.

BRB, due to its widespread availability and broad range of activities, seems to be a compound to which science should devote more attention. Of particular interest is the fact that this compound, among other anti-inflammatory activities, also exhibits a number of positive effects on the nervous system. This combination may be significant from the perspective of IBD therapy. Moreover, recent studies suggest that BRB is capable of delaying processes leading to neurodegenerative diseases [34,35,36]. This promising activity makes BRB a potential candidate for a new anti-inflammatory and immunomodulatory drug used for the treatment of IBD.

This review provides a comprehensive, cross-sectional study combining the molecular mechanisms of BRB’s anti-inflammatory action with comparisons of its efficacy to first-line drugs used in IBD (i.e., 5-aminosalicylic acid-5-ASA, sulfasalazine), as well as its effects on the gut microbiota, focusing on the upregulation and downregulation of specific bacterial species. In this way, it fills an information gap regarding the relationship between BRB-related changes and the course of gut–organ axis-dependent diseases, such as nonalcoholic fatty liver disease (NAFLD), ischemic stroke, and graft-versus-host disease (GVHD). Our main goal was to comprehensively present and summarize the current knowledge on the anti-inflammatory properties of this compound in the context of IBD treatment. In addition, we have demonstrated a number of clinical implications that support the continuation of research on this isoquinoline alkaloid as a potential anti-inflammatory and immunomodulatory agent.

## 2. Berberine Role in Colitis Alleviation

Inflammation of the intestines can be acute or chronic, and the distinction between the two is primarily related to the duration of the inflammatory process. Acute inflammation arises suddenly—usually within days—and resolves more quickly than chronic inflammation [37]. Etiological factors are most often viral, bacterial, or parasitic, in contrast to the causes of chronic inflammation, which are dominated by autoimmune diseases, long-term drug exposure, and parasitic infections [38]. The pathogenesis of acute inflammation involves neutrophils, cytokines, macrophages, and T and B lymphocytes [38]. In contrast, chronic inflammation involves T lymphocytes, B cells, dendritic cells, and neutrophils. Large amounts of cytokines are also produced, including IL-23, Th17 TNF, IL-1 β, and IFN-γ. They can result in the formation of strictures and even fistulas and abscesses, as observed in Crohn’s disease [39].

BRB has been extensively investigated for its anti-inflammatory properties. Multiple studies have employed in vivo models of colitis, including dextran sulfate sodium (DSS)—and trinitrobenzene sulfonic acid (TNBS)-induced colitis, as well as in vitro cellular assays, to elucidate BRB molecular and cellular mechanisms of action. Overall, BRB has consistently demonstrated protective effects against inflammation, epithelial damage, and oxidative stress, partly by modulating pro- and anti-inflammatory cytokines, enhancing epithelial barrier integrity, and interfering with key inflammatory signaling pathways.

Zhu et al. (2019) used an in vivo DSS-induced colitis model in rats to evaluate BRB effects [40]. BRB treatment reduced clinical symptoms such as weight loss, diarrhea, and colon shortening, producing outcomes comparable to sulfasalazine treatment. At the molecular level, BRB downregulated pro-inflammatory cytokines (IL-1, IL-1β, IL-6, IL-12, TNF-α, and IFN-γ) and upregulated anti-inflammatory cytokines (IL-4 and IL-10). Enzymatic activity associated with inflammation (iNOS, MPO) and oxidative damage (MDA) decreased. BRB also enhanced SIgA production and inhibited the IL-6/STAT3/NF-κB pathway, while increasing the expression of tight junction proteins (occludin, claudin, ZO-1, and VCAM-1), thereby improving epithelial barrier integrity [40].

Li et al. (2015) investigated BRB in an in vivo TNBS-induced colitis murine model [41]. Their results aligned with those of Zhu et al. (2019), showing reduced inflammatory damage in the large intestine and MPO activity [40]. BRB lowered the expression of pro-inflammatory cytokines (IL-1β, IL-6, IL-17, TNF-α, and IFN-γ) while increasing the anti-inflammatory cytokine IL-10. SIgA synthesis was also elevated. Furthermore, BRB modulated the immune response by promoting macrophage polarization toward the anti-inflammatory M2 phenotype and reducing M1 populations. It also inhibited Th1/Th17 differentiation via decreased STAT1/3 phosphorylation. Although BRB did not directly affect Treg cells, the reduced Th1/Th17 populations limited the overall pro-inflammatory activity [40].

Hong et al. (2012) and Yan et al. (2012) employed an in vivo DSS-induced colitis model in mice [42,43]. Hong et al. (2012) observed that BRB alleviated weight loss, bleeding, and diarrhea, decreased MPO activity, and downregulated pro-inflammatory cytokines (IL-12 and IFN-γ). Simultaneously, BRB upregulated IL-4 and IL-10 [42]. Yan et al. (2012) [43] similarly reported reduced weight loss and MPO activity alongside lowered pro-inflammatory cytokine levels (TNF-α, IFN-γ, KC, and IL-17). Macrophage apoptosis increased, stromal macrophage infiltration decreased, and TNF-α secretion by macrophages was suppressed. BRB also inhibited LPS-induced macrophage activation and apoptosis of colorectal epithelial cells. Key signaling pathways, including ERK1/2, p38, and NF-κB (via IκB stabilization), were inhibited, demonstrating robust anti-inflammatory effects.

Earlier research by Zhou et al. (2000) (in vivo TNBS-induced colitis in rats) and Lee et al. (2010) (in vivo TNBS-induced colitis in mice) further supported these findings [44,45]. Zhou et al. (2000) demonstrated that BRB reduced MPO activity and IL-8 secretion in colon epithelial and LPS-stimulated PBMC cells [44], while Lee et al. (2010) showed suppression of lipid peroxidation, weight loss, and MPO activity [45]. BRB downregulated pro-inflammatory cytokines (TNF-α, IL-1β, and IL-6) and partially restored IL-10 levels. Molecular analysis indicated reduced iNOS and COX-2 expression, silencing of the LPS/TLR4/NF-κB pathway, and finally the LPS-dependent MAPK pathway (ERK and JNK). These effects were often comparable or superior to sulfasalazine [45].

Li et al. (2016), utilizing an in vivo DSS-induced colitis model in mice, reported that BRB inhibited weight loss, improved stool consistency, and prevented colon shortening [46]. Molecular analyses revealed increased ZO1 and occludin expression, as well as improved epithelial barrier integrity. BRB reduced Th17 lymphocyte activity, IL-17 mRNA expression, and ROR-γt levels. Furthermore, it decreased pro-inflammatory cytokines (IL-6, IL-23, and TNF-α) and STAT3 pathway phosphorylation, leading to IL-17 silencing and suppression of colitis [46].

In another in vivo DSS-induced murine model, Liu et al. (2018) found that BRB modulated the AKT1/SOCS1/NF-κB signaling pathway to inhibit M1 macrophage polarization, and increased IL-10 secretion while decreasing IL-1β, IL-6, and TNF-α [47]. BRB also reduced intestinal mucosal infiltration by LPS-activated macrophages. Similarly, Zhang et al. (2017) demonstrated that BRB enhanced mucosal barrier function, upregulating epithelial barrier proteins (ZO-1, occludin, and E-cadherin), along with reduced MPO activity and enhanced catalase and peroxide dismutase activity [48]. BRB also downregulated pro-inflammatory cytokines, including TNF-α, IL-1β, and IL-6, and silenced STAT3 pathway phosphorylation, further reducing inflammation. Macrophage infiltration into the colon mucosa was significantly reduced, as evidenced by a decrease in CD68 expression in biopsy specimens. Li et al. (2018) confirmed these barrier-enhancing effects in a DSS-induced in vivo UC model, showing that BRB increased ZO-1, occludin, and claudin-1 levels. Moreover, BRB promoted mucosal regeneration, evidenced by increased expression of intestinal stem cell markers (LGR5 and TERT) [49].

Kawano et al. (2015), using an in vivo DSS-induced colitis model in C57BL/6 mice, reported that BRB reduced Th1/Th17 lymphocyte populations and the associated cytokines (TNF-α and IL-17), consistent with earlier studies [50]. BRB also antagonized dopamine D1 and D2L/S receptors, reducing lymphocyte secretion of pro-inflammatory cytokines. Furthermore, Li et al. (2017) demonstrated that BRB suppressed colitis-induced colorectal cancer progression in an in vivo DSS colitis murine model with Apc gene mutations, downregulating IL-6 and TNF-α and silencing the EGFR/ERK pathway [51]. BRB also reduced cytokine secretion in RAW264.7 macrophages, reinforcing its anti-inflammatory and anti-cancer potential.

In a more recent study, Li et al. (2024) employed an in vivo DSS-induced murine UC model and conducted molecular docking analyses [52]. BRB directly interacted with HIF-1α, TNF-α, and TLR4, disrupting their functions. Further analysis showed decreased mRNA expression of TLR4, NF-κB, and HIF-1α, with Western blot confirming silencing of the TLR4/NF-κB/HIF-1α signaling pathway. This disruption lowered cytokine secretion and alleviated clinical symptoms, such as weight loss, diarrhea, bleeding, reduced splenomegaly, and intestinal mucosal swelling, similar to mesalazine [52].

Zhai et al. (2020) used an in vivo DSS murine model to explore BRB molecular mechanisms [53]. BRB reduced lysophosphatidylcholine (LPC) formation and increased phosphatidylcholine (PC) by inhibiting phospholipase A2α activity, as confirmed by docking studies. Additionally, BRB-treated macrophages showed decreased gene expression of TNF-α, NOS2, IL-6, and CXCL10, correlating with phospholipase A2α suppression. This enzyme inhibition was identified as a key factor in BRB effectiveness against DSS-induced colitis and IBD. Furthermore, BRB downregulated the MAPK/JNK signaling pathway, indirectly reducing phospholipase A2α activity and resulting in significant inflammation reduction [53].

Zhou et al. (2020) investigated the effect of BRB on the circadian transcription factor Rev-Erbα and its relationship to the severity of DSS-induced chronic colitis in mice [54]. BRB was found to act as a Rev-Erbα agonist, reducing the expression of Bmal1 and Nlrp3, genes crucial to colitis pathogenesis. In a Rev-Erbα-dependent manner, BRB decreased the mRNA expression and synthesis of IL-1β, IL-6, Ccl2, Bmal1, and Nlrp3, demonstrating its direct anti-inflammatory effects. The dependence of BRB anti-inflammatory action on Rev-Erbα highlights the circadian (time-dependent) variability of its therapeutic effects. Accordingly, the anti-inflammatory effects of BRB were more pronounced when BRB was administered in the evening, aligning with the circadian activity of Rev-Erbα protein [54].

Wu et al. (2020) explored BRB capacity to restore Dicer gene expression in an in vivo model of pharmacologically induced colitis in heterozygous mice [55]. Dicer is crucial in maintaining intestinal homeostasis by modulating miRNAs involved in inflammation and epithelial barrier function. BRB enhanced Dicer expression and reduced IL-6 levels, mitigating inflammation and oxidative stress and thus maintaining intestinal homeostasis, similarly to anastrozole [55].

Finally, BRB anti-inflammatory potential was evaluated in humans with radiation-induced acute intestinal syndrome during oncological therapy [56]. Patients receiving BRB orally three times daily during pelvic or abdominal radiotherapy experienced a significant reduction in the frequency of radiation-induced intestinal syndrome (*p* < 0.05). This study highlights BRB clinical relevance and provides valuable insights into its therapeutic potential in human inflammatory conditions.

Collectively, these studies emphasize BRB anti-inflammatory potential. BRB modulates cytokine profiles by downregulating pro-inflammatory factors (e.g., TNF-α, IL-1β, IL-6, IL-8, IL-12, IL-17, and IFN-γ) and upregulating anti-inflammatory mediators (e.g., IL-4 and IL-10). It also promotes M2 macrophage polarization, limits Th1/Th17 responses, enhances epithelial tight junctions, and reduces oxidative stress. BRB interferes with multiple signaling pathways (e.g., TLR4/NF-κB, STAT3, MAPK, ERK/JNK, HIF-1α, and Rev-Erbα-related networks), leading to broad-spectrum anti-inflammatory and barrier-protective effects, as illustrated in Figure 2 and Figure 3.

### 2.1. BRB Derivatives

BRB derivatives have demonstrated significant anti-inflammatory effects in TNBS-induced colitis models. Demethylenberberine, a simple derivative of BRB, showed similar protective effects on the intestinal mucosa as BRB itself. Molecular analyses revealed that it silenced TLR4 signaling through both MyD88-dependent and -independent pathways, leading to reduced expression of key inflammatory markers, including IL-6, IL-1β, TNF-α, IFN-γ, TRAF6, and IRF3. Additionally, the NF-κB signaling cascade was inhibited by reducing IκB phosphorylation. Both in vitro and in vivo studies with LPS-stimulated RAW264.7 macrophages demonstrated that demethylenberberine and BRB reduced the expression and secretion of these cytokines. Docking studies also confirmed that demethylenberberine binds to the MD-2 protein domain, directly disrupting LPS-TLR4 interactions. This mechanism significantly attenuated the inflammatory response and increased survival in a mouse model of LPS-induced sepsis [57].

Zhang et al. (2016) further explored new BRB derivatives, including (±)-8-acylmethyldihydroberberine-type and tertiary dihydroberberine-type alkaloids [58]. These compounds, tested in vitro, induced the expression of XBP1, an immunomodulatory protein crucial for mitigating inflammation. These findings highlight the therapeutic potential of BRB derivatives in treating inflammatory diseases such as UC [58].

### 2.2. BRB in Combination with Other Compounds

The anti-inflammatory effects of BRB in combination with other compounds have been evaluated in various in vivo models, demonstrating enhanced efficacy and potential synergistic mechanisms. For example, Zhang et al. (2011) investigated BRB, hypaconitin, and skimmianin in an in vivo TNBS-induced rat model of UC [59]. BRB exhibited the strongest protective effect on colon function, including histological improvements in epithelial structure and reduced macrophage infiltration. These beneficial outcomes were further enhanced when all three alkaloids were combined. In LPS-stimulated macrophages, only BRB alone downregulated TNF-α secretion, but combining the three alkaloids reduced the effective dose of BRB needed to achieve its anti-inflammatory action. Similarly, the combination lowered LBP secretion by immune cells, an effect not observed with BRB alone. Notably, the three alkaloids together showed stronger inhibition of the LPS/TLR4/NF-κB pathway compared to individual compounds, further highlighting the synergistic interaction [59].

Li et al. (2015) reported synergistic effects of BRB combined with 5-aminosalicylic acid (5-ASA) in a DSS-induced colitis model in C57BL/6 mice [60]. The combination therapy outperformed 5-ASA alone in improving stool consistency, reducing rectal bleeding, and alleviating inflammation, as evidenced by histological evaluation. Although no significant improvement in intestinal shortening was observed, the combined treatment more effectively downregulated pro-inflammatory markers, including IL-6, IL-23, IL-12b, TNF-α, and COX-2. Western blot analysis showed suppression of the NF-κB and JAK2/STAT signaling pathways, correlating with reduced TNF-α secretion by murine lymphocytes [60].

In a murine in vivo colitis model, Jeong et al. (2014) evaluated a BRB-rich rhizome mixture of *Anemarrhena asphodeloides* and *Coptidis chinensis* [61]. Compared to BRB alone, the mixture more effectively reduced weight loss and intestinal shortening, inhibited MPO activity, and downregulated pro-inflammatory cytokines (IL-6, IL-1β, and TNF-α), while upregulating the anti-inflammatory cytokine IL-10. At the molecular level, the mixture decreased iNOS and COX-2 expression and suppressed NF-κB signaling. In in vitro assays using LPS-activated macrophages, the mixture inhibited IRAK1 and IKK phosphorylation and silenced NF-κB activity, reducing cytokine secretion. Notably, BRB directly interfered with LPS-TLR4 interactions, contributing to the mixture’s overall anti-inflammatory effect [61].

Liu et al. (2021) investigated BRB contained in the *Coptidis rhizoma* extract in a DSS-induced murine colitis model [62]. The combination of BRB with the extract outperformed BRB alone, alleviating inflammation by decreasing MPO activity and downregulating IL-1β, IL-6, TNF-α, Ccl-2, and Nlrp3 expression. Interestingly, the anti-inflammatory effects were more pronounced when the compounds were administered in the evening, aligning with the circadian activity of Rev-Erbα protein. This study underscores the potential importance of timing in BRB administration [62].

In vivo models of subchronic and chronic UC in mice showed that BRB combined with fingolimod (a treatment for multiple sclerosis) more effectively reduced weight loss, bleeding, and intestinal shortening than BRB alone. The combination also downregulated IL-6, IL-17, and TNF-α expression. Molecular analysis revealed attenuation of the IL-6, indicating synergistic anti-inflammatory effects [63].

Gao et al. (2023) found that BRB and hesperetin-based nanoparticles showed higher anti-inflammatory potential than BRB or hesperetin alone [64]. In an in vivo model of UC induced via DSS in mice, the treatment of BRB with nanocomplexes resulted in reduced weight loss, bleeding, and diarrhea while downregulating TNF-α and IL-6 and upregulating IL-10 levels. Key intestinal mucosal proteins: ZO1, claudin-1, and occludins were also upregulated. The nanoparticles further reduced infiltration of neutrophils (CD45) and Tγδ lymphocytes, indicating immunosuppressive properties. At the molecular level, Western blot analysis showed that nanoparticle exposure suppressed the PI3K/Akt signaling pathway via dephosphorylation [64].

The anti-inflammatory effects of Gegen Qinlian decoction (high BRB content) compared to BRB alone were evaluated in a DSS-induced in vivo colitis model in mice. Both treatments yielded similar benefits, including reduced colon shortening, weight loss, and diarrhea, alongside histopathological improvements in mucosal integrity. Both suppressed the TLR4/NF-κB pathway, evidenced by decreased levels of p-p65 and p-IκB, and secretion of pro-inflammatory cytokines (TNF-α, IL-1β, IL-4, and IL-6). Importantly, the decoction and BRB also modulated the in vitro activity of RAW 264.7 macrophages in the LPS activation test, interfering with macrophage activation through the TLR4/NF-κB pathway. Additionally, oxidative stress was mitigated by elevated reduced glutathione levels and decreased MPO activity [65].

### 2.3. BRB Advanced Delivery Systems

Advanced delivery systems have been used for BRB to enhance its bioavailability and therapeutic potential in inflammatory conditions. Ding et al. (2020) demonstrated the efficacy of BRB encapsulated in Konjac Glucomannan Hydrogel in a DSS-induced UC model (murine model in vivo) [66]. Compared to free BRB, hydrogel-encapsulated BRB significantly enhanced the reduction in weight loss and colorectal shortening. Molecular analyses also revealed a greater decrease in mRNA expression and protein levels of TNF-α, IL-6, and IL-1β, along with reductions in MPO activity and ROS formation, indicating a superior anti-inflammatory profile [66].

BRB in lipid nanoparticles was evaluated for anti-inflammatory properties in a murine model of DSS-induced colitis by Deng et al. (2020) [67]. Improved uptake by RAW 264.7 macrophages and Caco-2 colonocytes was observed in vitro. In vivo, the nanoparticles reduced intestinal shortening, improved histological condition, and alleviated diarrhea and bleeding. At the molecular level, BRB upregulated ZO1 expression and downregulated mRNA levels of IL-1β, IL-6, MMP-9, CX3CR1, COX-2, and TERT. Western blot analysis showed inhibition of NF-κB signaling by blocking the nuclear translocation of the activated factor [67].

Feng et al. (2022) explored the anti-inflammatory effects of BRB and epigallocatechin gallate encapsulated in yeast microcapsules in a DSS-induced colitis model [68]. The yeast antigens facilitated dectin-1-mediated macrophage uptake, polarizing them toward an anti-inflammatory M2 phenotype. The treatment significantly reduced TNF-α, IL-1β, H_2_O_2_, and malondialdehyde levels, indicating reduced cytokine secretion and oxidative stress compared to BRB alone and BRB nanoparticles [68].

Similarly, Xu et al. (2022) investigated BRB nanoparticles with a fungal antigen (β-glucan) in a DSS-induced murine colitis model [69]. BRB nanoparticles with β-glucan were also tested in vitro against LPS-activated RAW 264.7 macrophages and Caco-2 cells. Efficient macrophage uptake led to reduced expression of IL-6 and IL-1β, as well as decreased COX-2 and iNOS activity. Histological analysis revealed decreased macrophage infiltration into the colon mucosa and improved intestinal tissue condition (compared to the model with BRB alone) [69].

Zhang et al. (2021) evaluated micro- and nano-encapsulated hybrid delivery systems containing BRB in DSS-induced colitis [70]. A single dose of the BRB nano-system was more effective than weekly free BRB therapy in acute colitis, preventing further weight loss. In chronic colitis, the nano-system better reduced splenomegaly and downregulated IL-1β, TNF-α, and IL-6 expression compared to free BRB [70].

The advanced delivery systems, including hydrogels, nanoparticles, and microcapsules, have demonstrated significant improvements in BRB stability, cellular uptake, and targeted delivery. These innovations not only enhance BRB anti-inflammatory effects but also provide promising strategies for more effective treatment of inflammatory conditions.

## 3. Berberine—Effect on the Development and the Course of Bacterial Infections

After oral administration, BRB predominantly remains within the intestinal lumen, where it undergoes metabolism by host and microbiota enzymes. This results in the local action of the alkaloid in the gastrointestinal tract while maintaining low systemic bioavailability [71,72]. Through mechanisms similar to those involved in the alleviation of DSS-induced colitis described above, BRB may play a key role in preventing various types of intestinal infections, including intoxications, toxicoinfections, and invasive infections.

### 3.1. Effect on Toxic Bacterial Products

The antisecretory and antidiarrheal properties of BRB have been recognized in traditional Chinese and Indian medicine for centuries. Experimental studies using rabbit ligated intestinal loop models have demonstrated that BRB inhibits the intestinal secretory response induced by heat-labile *Vibrio cholerae* enterotoxins and *E. coli* toxins by approximately 70%. This effect was observed both when BBR was administered before and after toxin exposure, indicating modulation of secondary secretory mechanisms rather than direct inhibition of adenylyl cyclase activity [73]. Further research conducted by Zhang et al. (2012) showed that BBR acts by modifying ion transport through the regulation of intestinal epithelial transporters and channels, notably NHE3 and AQP4 [74].

Additionally, BRB protects the intestinal barrier integrity and indirectly reduces excessive activation of pattern recognition receptors (e.g., TLR4) and downstream NF-κB signaling following LPS exposure. In studies by Zhu et al. (2020) using porcine models of intestinal epithelial cells, pre-treatment with BRB reduced the expression of IL-1β, IL-6, TNF-α, and other pro-inflammatory mediators, resulting in decreased epithelial damage and reduced secretion of water and electrolytes into the intestinal lumen [75].

### 3.2. Inhibition of Toxicoinfections

In toxicoinfections, pathogens adhere to and colonize the intestinal epithelium, secrete toxins, and frequently form biofilms. In this case, BRB’s actions are diverse and multifaceted.

**Inhibition of bacterial growth, colonization, and their antibiotic resistance:** In in vitro studies, BRB demonstrates bacteriostatic or even bactericidal activity against numerous, often multidrug-resistant (MDR), enteropathogens. For example, BBR damages bacterial cell walls and membranes by reducing the antioxidant capacity of *Staphylococcus aureus* and promoting the accumulation of D-Ala-D-Ala precursors in MRSA (methicillin-resistant *S. aureus*) strains [76,77]. Moreover, it disrupts nucleic acid function and inhibits metabolism and motility processes in *E. coli* [77]. In *Pseudomonas aeruginosa*, BRB functions as an efflux pump inhibitor, reducing MexXY-dependent resistance to multiple antibiotics (e.g., cephalosporins, aminoglycosides, macrolides, lincosamides) [32]. Consequently, BRB acts synergistically with several antibiotic classes (e.g., ciprofloxacin, tobramycin) to inhibit bacterial growth and colonization, as confirmed in in vitro studies [78].

**Inhibition of adhesion and biofilm formation:** BRB reduces the synthesis and expression of fimbrial genes and other adhesion-associated factors, e.g., type I fimbriae in *Salmonella enterica* subsp. *enterica* serovar Typhimurium [79]. This limits bacterial adherence and colonization of the intestinal epithelium by pathogenic bacteria. Sub-inhibitory concentrations of BRB also disrupt quorum-sensing (QS) systems—cell-to-cell communication mechanisms essential for biofilm formation of invasive pathogens [80].

**Indirect effects via the gut microbiota:** BRB alters gut microbial composition by suppressing opportunistic pathogens and enhancing colonization resistance against enteric pathogens. These interactions are discussed in detail later in this article (for details, see chapter 4).

### 3.3. Invasive Infections

In invasive infections, pathogens not only colonize the intestinal surface but also penetrate epithelial layers, disseminate into local tissues and blood vessels, and may even enter systemic circulation (e.g., *Salmonella* spp., *Shigella* spp., *Campylobacter* spp.). Consequently, BRB’s action extends beyond the intestinal lumen and includes direct inhibition of bacterial virulence, as well as indirect strengthening of the intestinal barrier and systemic modulation of the immune response.

**Reduction in bacterial virulence:** Choi et al. (2023) reported that BRB inhibits the type II secretion system (TTSS) of *S*. *typhimurium*, leading to decreased expression of invasion and intracellular survival genes [81]. Similarly, Fu et al. (2010) demonstrated that BRB modulates the expression of genes associated with secretion and virulence of *Shigella flexneri*, suggesting a potential reduction in its invasive capacity [82]. In a recent in vitro model of *Campylobacter jejuni* infection, Duda-Madej et al. (2025) demonstrated that BRB (MIC = 64 µg/mL) effectively limits bacterial growth and biofilm formation while protecting colonocytes from toxic metabolites and intercellular junction disruption [83]. These findings indicate that BRB simultaneously attenuates *C. jejuni* virulence and stabilizes the intestinal barrier, thereby mitigating the consequences of invasive infection [83]. Although, to the best of our knowledge, no direct studies have yet explored the effects of BRB on *Campylobacter* infection in vivo, parallels drawn from studies on other invasive pathogens highlight its potential therapeutic relevance in controlling campylobacteriosis. Further investigation in this field remains essential, as the treatment of invasive bacterial infection is increasingly challenging due to the global rise in multidrug-resistant (MDR) strains. According to the World Health Organization (WHO), antibiotic resistance among invasive pathogens—including *Salmonella* spp. and *Shigella* spp.—increased by more than 40% between 2018 and 2023 across monitored pathogen-antibiotic combinations [84]. *Campylobacter* strains also rapidly acquire resistance. This alarming situation, together with further studies, is warranted to elucidate the molecular mechanisms underlying BRB’s antimicrobial and barrier-protective effects, and to evaluate its potential as a therapeutic adjunct in the management of campylobacteriosis [85].

**Strengthening the intestinal barrier:** In a murine endotoxemia model (10 mg/kg LPS intraperitoneally), BRB enhanced the expression of tight junction proteins, i.e., occludin, claudin-1, and ZO-1. In this way, BRB prevented pathological disruption of epithelial integrity and limited bacterial translocation from the intestinal lumen to the systemic circulation [86].

**Systemic immunomodulation:** In invasive models, BRB inhibits pro-inflammatory signaling pathways (TLR4/NF-κB and MAPK), suppresses cytokine production (IL-1β, IL-6, and TNF-α), and consequently reduces leukocyte migration (for details, see chapter 2). These effects collectively mitigate excessive inflammation during invasive infections, thereby reducing tissue injury and preventing bacterial dissemination [87].

## 4. Berberine Modulating Effect on Gut Microbiota via Intestinal Barrier

### 4.1. Direct Modulation

#### 4.1.1. Impact on Mucin-Secreting Goblet Cells

BRB exerts beneficial effects on the intestinal environment, in part through its action on mucin-secreting cells. This occurs via two complementary mechanisms: (i) regulation of mucin levels, which enhances its protective, mucolytic, and/or anti-inflammatory functions [88,89], and (ii) an increase in the number of goblet cells driven by modulation of their proliferation [90]. From the perspective of the present study, the second mechanism is of particular relevance. Through this pathway, BRB leads to: (i) an increased number of goblet cells and thickening of the mucus layer in the small intestine [91]; (ii) expansion of mucous glands in the colon [92], and (iii) modulation of the gut microbiota, including an increase in *Akkermansia muciniphila*, and SCFA-producing bacteria such as *Agathobacter, Bifidobacterium*, and *Holdemanella*, along with a reduction in pathogenic taxa such as *Prevotella, Collinsella, Catenisphaera*, and *Enterobacter* [90,91]. The effect of BRB on goblet cells is shown in Figure 4.

#### 4.1.2. Intestinal Disorders

BRB is increasingly recognized not only for its direct immunomodulatory and anti-inflammatory properties (discussed in detail in Chapter 2, using studies in a mouse model of DSS-induced UC) but also for its ability to modulate the gut–microbial axis. An increasing number of studies highlight the particularly important role of BRB in mitigating intestinal dysbiosis, and consequently in restoring proper synthesis of microbiota-derived compounds (e.g., SCFAs, bile acid derivatives, indoles, tryptophan degradation products), as well as in reestablishing the integrity of the intestinal barrier previously damaged in the course of UC.

Taxonomic profiling by Yu et al. (2024) revealed that BRB increased the proportion of *Bacteroidetes*, decreased *Proteobacteria*, and regulated genera such as *Muribaculaceae* and *Turicibacter* [93]. *Bacteroidota*, dominant in the intestinal microbiome, play a crucial role in producing short-chain fatty acids, which are significantly reduced in active inflammatory bowel disease. Conversely, *Proteobacteria* increase in colitis patients and are associated with inflammatory conditions [94]. An increase in *Muribaculum* (a genus within *Bacteroidota*) has been linked to alleviating colitis-associated carcinogenesis, while *Turicibacter* is associated with exacerbating inflammation [95].

Xu et al. (2023) [96] also explored the synergistic effects of berberine hydrochloride (BBH) and dehydrocostus lactone, finding that BBH restored microbiota composition in DSS-induced UC models compared to that of untreated and healthy mice. BBH increased *Bacteroidetes* and *Verrucomicrobiota* while reducing *Firmicutes*. It significantly enhanced beneficial bacteria, such as *Verrucomicobiota* and its offshoot genus *Akkermansia*. At the same time, microbiota changes following BRB administration are associated with improved bacterial metabolism, thereby preventing and alleviating gastrointestinal disorders (IBS, IBD), as well as metabolic diseases (type II diabetes, obesity, and insulin resistance). The studies cited below demonstrate that BRB affects not only the composition of the microbiome but also its functional remodeling.

X. Sun et al. (2023) [97] assessed microbiota metabolites, such as short-chain fatty acids, bile acids, 5-hydroxytryptamine, and others, noting that BRB reversed their reductions caused by DSS. Additionally, BRB treatment increased *Firmicutes* and reduced *Bacteroidetes*, although these changes did not reach statistical significance. At the family level, BRB reversed the DSS-induced reduction in *Lactobacillus*.

Zhang et al. (2012) [98], in turn, conducted a detailed analysis of 16S rRNA in the intestines of rats. They demonstrated that treatment with BRB and metformin increased the abundance of SCFA-producing strains—including the genera *Blautia* and *Allobaculum*—which exerted beneficial effects in preventing obesity and insulin resistance in the studied rat groups.

In rat models of IBD, BRB regulated tryptophan metabolites, enhancing intestinal flora diversity and increasing *Lactobacillus*, *Clostridium*, and *Akkermansia*. Tryptophan metabolites activate the aryl hydrocarbon receptor (AhR), which positively impacts the intestinal epithelial barrier [99].

Gegen Qinlian Decoction, which contains BRB as a major component, upregulated *Faecalibacterium* and *Roseburia* (butyrate-producing microorganisms) in diabetic rats, thereby increasing fecal short-chain fatty acids levels [100].

In a murine model of irritable bowel syndrome (IBS), BRB combined with baicalin nanoparticles produced greater therapeutic effects than either compound alone. The combination altered intestinal flora (*Cyanobacteria, Bacteroidia, Verrucomicrobia, Deferribacteres, Candidatus,* and also *Saccharibacteria*) and reduced NF-κB signaling, along with decreasing levels of vasoactive intestinal polypeptide, serotonin, and choline acetyltransferase [101].

In type II diabetic mice, combining BRB with starches enhanced the intestinal flora composition. This combination significantly increased *Akkermansiaceae* and *Lactobacillus* populations compared to BRB alone, which otherwise reduced *Lactobacillus* levels [102].

Studies on BRB metabolism indicate that bacterial enzymes, such as CYP51 (produced by *E. coli*), convert BRB into metabolites like M1-talifendine and M2-berberberine. Inhibition of this process by voriconazole increases BRB intestinal absorption [70]. Importantly, BRB prevents endotoxin entry into the bloodstream, increasing short-chain-fatty-acid-producing bacteria that support mucosal integrity and reduce inflammation [103].

In summary, by influencing the microbiota and its metabolism, BRB may indirectly support intestinal homeostasis, promote the production of beneficial metabolites, and improve the intestinal environment. This represents an important component of its protective effect against infection and inflammation.

#### 4.1.3. Liver Disorders

Wang et al. (2024) [104] explored the effects of BRB on **cholestatic liver injury** in primary sclerosing cholangitis (PSC), demonstrating its ability to improve intestinal barrier function, reduce bacterial translocation, and restore gut microbiota and bile acid homeostasis. Using an in vivo PSC model, BRB improved the gut–liver axis by increasing *Bacteroidetes* (69% vs. 65% in control) and reducing *Firmicutes* (30% vs. 34%). In this way, it indirectly played a key role in correcting dysbiosis, modulating metabolic and inflammatory factors, and strengthening the intestinal barrier [105]. At the same time, through changes in *Firmicutes*, it reduced the activity of bile salt hydrolase (BSH), leading to a reduction in harmful secondary bile acids, such as deoxycholic acid (DCA) and lithocholic acid (LCA) [106]. BRB also reduced inflammation by downregulating MCP-1, CD11b, IL-1β, VCAM-1, IL-1α, and CXCL1, and inhibiting NF-κB and MAPK signaling pathways [104].

Zhang (2018) and Dai (2022) evaluated the combined effects of BRB and evodiamine in an in vivo NAFLD model [107,108]. BRB and evodiamine increased beneficial gut bacteria (e.g., *Ruminococcus* spp., *Lactobacillus* spp., *Prevotella* spp.) while reducing pathogenic genera (*Lachnospira* spp., *Fusobacterium* spp.). Enhanced intestinal barrier function and reduced apoptosis of intestinal epithelial cells were observed, along with decreased production of pro-inflammatory factors [107].

Guo et al. (2023) investigated the therapeutic effects of BRB and MTF in HFD-fed hamsters and ApoE(−/−) mice, demonstrating similar efficacy in reducing fatty liver, atherosclerosis, and inflammation [109]. BRB was more effective in reducing obesity and hyperlipidemia, while MTF was better at controlling blood glucose. 16S rRNA sequencing revealed that MTF increased *Bifidobacterium* and *Lactobacillus*, correlating with improved hyperglycemia, whereas BRB increased *Phascolarctobacterium* and *Blautia*, linked to reduced hyperlipidemia. Both treatments elevated *Bacteroides* and *Akkermansia*, associated with improved hepatic steatosis and atherosclerosis. Additionally, both drugs reduced pro-inflammatory cytokines (TNF-α, IL-6, IL-1β, and NF-κB), highlighting their anti-inflammatory and metabolic benefits [109].

Studies by Xu et al. (2017) on rats fed a high-fat diet showed that supplementation with BRB reduced intestinal barrier permeability, preventing bacterial LPS from entering the blood vessels and thereby inhibiting the inflammatory cascade [110]. In addition, glucagon-like peptides 1 and 2 and peptide YY were improved in the presence of BRB. Markers of infiltrating macrophages (MCP-1, F4/80), inflammation (IL-1β, PAI-1), and oxidative stress (STAMP-2, NADPHox) were also reduced [110].

Cao et al. (2016) studied BRB in mice with **nonalcoholic steatohepatitis**, revealing an increase in *Bifidobacterium* and a higher *Bacteroidetes/Firmicutes* ratio [111]. These changes correlated with reduced body weight, glucose, lipids, and insulin levels, alongside decreased pro-inflammatory markers (IL-6, TNF-α, IL-1β, and CD14) [111].

Chen et al. (2023) investigated the effects of BRB and metformin (MTF) in rats with nonalcoholic steatohepatitis [112]. Both treatments reduced hepatic steatosis, inflammation by alleviating damage to the intestinal mucosa and gut microbiota disruption. They specifically restored the diversity of the intestinal microbiota and altered the abundance of bacterial groups such as *Atopobiaceae*, *Christensenellaceae*, and *Rikenellaceae*, highlighting its role in modulating the gut microbiome [112].

#### 4.1.4. The Effect of Berberine on the Bacteroidetes/Firmicutes Ratio

*Firmicutes* and *Bacteroidetes* constitute more than 90% of human gut microbiota, and the balance between these two phyla has been associated with several pathological conditions [113]. BRB has been shown to modulate gut microbial composition; however, its impact on the *Firmicutes/Bacteroidetes* ratio is highly context-dependent, varying with the disease model, dosage, and duration of treatment. Several studies indicate that BRB primarily reverses disease-induced dysbiosis and promotes the re-establishment of microbial community structure resembling that of healthy individuals [93,97,104,111,114]. In inflammatory models, such as DSS-induced colitis, BRB contributes mainly to a broader normalization of dysbiosis, rather than a consistent and unidirectional shift in either phylum [93].

Collectively, these observations indicate that BRB supports the restoration of gut microbiota equilibrium. However, its specific effects are context-dependent and should not be generalized as a universal pattern. Further research is needed to clarify the specific impact of BRB on the *Bacteroidetes/Firmicutes* balance across different disease settings.

### 4.2. Indirect Modulation

According to recent data, age-related changes in gut microbiota composition, along with immune imbalance, may lead to chronic neuroinflammation and blood–brain barrier disruption, contributing to cognitive decline and the development of neurodegenerative diseases (e.g., dementia, AD). Interventions that restore a healthy microbiota and immune balance—such as gut flora modulation—represent a promising strategy to protect against cognitive decline in aging [115].

One such intervention is BRB, which, by modulating the composition of the gut microbiota, improving the integrity of the gut barrier, and inhibiting chronic immune activation, may indirectly limit the neuroinflammatory processes that contribute to the development of cognitive impairment and dementia.

For example, C. Sun et al. explored the relation of central nervous system disorders, such as **Alzheimer’s disease (AD)**, to the intestinal flora and its metabolism [116]. The state of the gut microbiota influences the improvement of neurological diseases through two-way signaling in the gut–brain axis. Dysfunction of the gut–blood barrier leads to increased penetration of small molecules such as amyloids, LPS-induced cytokines, as well as other pro-inflammatory mediators. When the composition of the microflora is disrupted, an increase in bacterial products such as metabolites and proteins can happen, which can lead to increased amyloid beta aggregation and Tau protein folding [16]. In a study by Sun et al. (2024), BRB treatment mitigated intestinal permeability through upregulation of occludin and claudin proteins, preventing the aforementioned pathological processes [116]. BRB treatment also led to the clearance of the amyloid plaques, reduction in neuroinflammation, and improved spatial memory problems. In order to assess the diversity of gut microbiota, murine feces were subjected to 16S rRNA sequencing, and the analysis showed that *Enterococcus, Ligilactobacillus, Akkermansia, Roseburia*, and *Lachnoclostridium* in the AD model group (5xFAD model mice) were found to be low in abundance, while the BRB treatment reversed this trend. *Paraprevotella*, *Alloprevotella*, and *Bacteroides* were abundant in the AD model, and following BRB treatment, the species abundance significantly decreased. The integrity of the intestinal barrier was also evaluated, and the findings suggested that BRB alleviates intestinal permeability through upregulation of occludin and claudin proteins [116].

Duan et al. (2023) explored how butyrate, a product of gut microbiota, ameliorates ischemic stroke, and how BRB mediates its production [117]. The transplantation of gut microbiota previously treated with BRB considerably improved stroke outcomes in mice compared to the group in which the saline-treated microbiome was transplanted. To further investigate the role of butyrate in this process, an inhibitor of butyrate synthesis, heptanyl-coenzyme A, was administered, which caused a reversal of the ameliorating effect of BRB. The study indicates that BRB modulates the gut microbiome and promotes the production of butyrate to improve ischemic stroke outcome in mice [117].

Chen et al. (2023) reported anti-tumor effects of BRB in high-fat diet (HFD)-associated colorectal cancer and its role in modulating gut microbiota [118]. The use of BRB reduced the number of colonic polyps, lessened intestinal barrier disruption, and inhibited intestinal inflammation in AOM(Azoxymethane)/DSS-induced colorectal mouse cancer model. BRB was found to alleviate microbiota dysbiosis in mice fed with HFD and caused increased abundance of beneficial gut microorganisms such as *Akkermansia* and *Parabacteroides,* while decreasing potentially pathogenic bacteria, including *Alistipes* and *Clostridium sensu stricto 1*. Interestingly, BRB had no clear anti-tumor effect in mice with depleted gut microbiota, while transplantation of gut microbiota from mice treated with BRB to mice with depleted microbiota restored the inhibitory effect on tumorigenesis [118]. Research by Wang et al. (2021) supplemented these observations [119]. They assessed the effects of BRB on occludin and ZO-1—proteins involved in intestinal epithelial tight junctions—as well as on mucins in rats with prediabetes. Increased mucin levels were noted in BRB-treated rats, along with upregulation of occludin and ZO-1. Inflammation was also reduced, while plasma GLP-2 levels and glutamine-induced intestinal secretion increased [119]. Pan et al. (2022), in turn, provided valuable findings on the protective effect of BRB treatment on sepsis associated with gut microbiota using CLP (cecal ligature and puncture) rats [120]. BRB mitigated the histological injury of the ileum and inhibited interleukin IL-1β, IL-6, IL-17A, and monocyte chemokine-1 serum levels. 16S rDNA sequencing analysis showed that CLP rats had an altered abundance of gut microbiota compared to control animals, with *Lactobacillus* being the most significantly increased in the CLP group. BRB treatment resulted in the reversion of microbiota disruption and caused the microbiota composition to be closer to that of the control group. Additionally, the results indicated that BRB can improve intestinal barrier function disorders [120].

The anti-inflammatory effect of BRB in **acute graft-versus-host disease (aGVHD**) was evaluated by Zhao et al. (2022) [121]. To induce aGVHD, female BALB/c and male C57BL/6 mice underwent intravenous injections of donor bone marrow cells combined with splenocytes. The results showed that gavage administration of BRB alleviated GVHD-induced weight loss, high mortality, and inhibited inflammation and organ damage, as well as prevented destruction of the intestinal barrier. BRB also worked as an inhibitor of NLRP3 (nucleotide-binding domain and leucine-rich repeat protein-3) inflammasome, a complex that controls the activation of pro-inflammatory cytokines. The microbiota changes in GVHD mice were analyzed, and BRB showed potential to renovate the abundances of genus *Adlercreutzia, Dorea, Sutterella,* and *Plesiomonas.* Additionally, BRB treatment could cause an increase in Lactobacillus abundance in GVHD mice [121]. Another study performed by Zhu et al. (2018) on ApoE−/− mice (mice with a deficiency of apolipoprotein E) fed a high-fat diet treated with BRB showed an increase in the number of the *Akkermansia* bacterial strain, a decrease in the level of pro-inflammatory cytokines (IL-1β, TNF-α, and IL-6), and a strengthening of the intestinal barrier by increasing the activity of integral proteins (ZO-1), which was also confirmed in previous studies [122]. A similar research was conducted by Wu et al. (2020) on ApoE−/− mice fed a high-fat diet [55]. An anti-inflammatory effect was achieved, manifested by a decrease in pro-inflammatory cytokines IL-1β, IL-6, and TNF-α. On the other hand, the concentration of anti-inflammatory cytokines such as adiponectin and IL-10 increased. Significant changes also occurred in the intestinal flora of mice, in which the number of bacteria from the strains *Alistipes, Bilophila, Roseburia Blautia, Turicibacter, and Allobaculum* increased. Importantly, these changes resulted in the increase in the bacterial production of SCFA [55]. SCFAs constitute a large group of organic acids (e.g., butyric acid), referred to as postbiotics—microbiome-derived metabolites beneficial in IBD therapy [123]. These substances exhibit pleiotropic protective mechanisms by influencing multiple inflammatory pathways, including supporting mitochondrial respiration, ATP synthesis, and anti-oxidative effect in the area of colonocytes. Moreover, it has been reported that SCFAs can contribute to the limitation of the colonocytes apoptosis, which allows for the maintenance of intestinal barrier integrity and thus reduces intestinal damage in the inflammatory process [123].

An in vivo mouse model of DSS-induced colitis was also used in the study by Li et al. (2020), who tested the anti-inflammatory properties of oxyberberine [101]. Importantly, this compound reduced the amount of unfavorable intestinal flora, which includes, among others, *Paraprevotella, Enterococcus,* and *Ruminococcus*. Interestingly, oxyberberine was also used in studies by Li, Q.P et al. (2021) on an obese murine model in vivo with NAFLD at a dose of 100 mg/kg; the inflammation of adipocytes was reduced by acting on the mRNA MCP-1, CD11c, CD68, and NOS2 [124].

The summary of the BRB modulating effect on gut microbiota is presented in Figure 5.

### 4.3. Nanoparticles in Addition to Berberine

An interesting combination is carboxymethyl chitosan nanochitosan particles with the addition of BRB, whose effects were examined in a murine model of inflammatory bowel disease. Colitis was induced by the administration of DSS. A reduction in colonic inflammation was observed, along with an increase in the diversity of beneficial intestinal flora. Inhibition of IL-6 production was also noted [125].

In summary, the gut–immune axis works bidirectionally in systemic diseases. On the one hand, changes in the microbiota and intestinal barrier can lead to the penetration of microbiota or its products (endotoxins, metabolites) into circulation, triggering inflammation even in distant organs (eg., the central nervous system). On the other hand, systemic inflammation, infections, or immune stress can disrupt the microbiota and intestinal barrier—creating a cycle that exacerbates dysfunction.

By modulating the microbiota, metabolites, and stabilizing the intestinal barrier, BRB can interrupt this pathological cycle, making it a promising intervention not only for intestinal infections but also for systemic diseases whose pathogenesis is linked to disruption of the gut–immune axis.

The summary of the BRB impact on the mentioned internal and systemic diseases is presented in Table 1.

## 5. Limitations of the Evidence Presented

Despite the large number of literature references cited in this review, several methodological and interpretative limitations cannot be overlooked. The use of different DSS/TNBS protocols with varying disease severity makes it difficult to compare results between studies from different research centers and may contribute to differences in the conclusions drawn. In addition, small study samples, short treatment times, and the lack of in-depth microbiome analyses do not allow for unequivocal confirmation of the effectiveness of BRB. It is also worth noting that some studies focused on determining the levels of cytokines, oxidative stress markers, or intestinal barrier proteins without correlating them with the clinical picture, i.e., remission of inflammation or reduction in symptom severity. Furthermore, the *Firmicutes/Bacteroidetes* ratio remains controversial as it is highly dependent on dietary differences, inflammation levels, intestinal barrier integrity, and metabolic differences. Therefore, studying these correlations without focusing on specific mechanisms is highly inconsistent. These numerous limitations of the data available in the literature further emphasize the need to expand and refine research on the impact of BRB on the intestinal environment.

## 6. Research Gaps in the Field of Berberine

Although it is known that BRB strengthens tight junctions, improves mucosal integrity, and influences goblet cell differentiation, many of its molecular mechanisms remain unexplained. The major obstacle to further research is its low bioavailability and the lack of safety assessment of advanced delivery systems. In particular, there is an information gap regarding their interaction with the immune system and microbiota. Therefore, comprehensive scientific research combining metagenomics, metabolomics, transcriptomics, and proteomics is necessary. Such a multi-omic approach will undoubtedly help to understand the action of BRB and enable the determination of the therapeutic window and optimization of the dose of this alkaloid.

## 7. Conclusions

Inflammatory bowel diseases are known to influence cancer progression, particularly in the colon. Conditions like Crohn’s disease can lead to malabsorption, and complications may include gastrointestinal perforation, bleeding, peritonitis, and even death. Due to the potential for severe outcomes such as intestinal stenosis, the development of fistulas or inflammatory tumors, rapid treatment aimed at alleviation of inflammation is now a priority. Unfortunately, many effective drugs for these diseases come with significant side effects. Immunosuppression-based therapy (including biologics, e.g., infliximab, etanercept; calcineurin inhibitors—cyclosporin) increases the risk of opportunistic infections or secondary cancers. This has led to rising interest in immunomodulatory drugs, which aim to polarize the immune response and minimize inflammatory damage to tissues.

Substances of natural origin, often known from traditional medicine, have shown potential in treating these diseases. The described alkaloid BRB has demonstrated anti-inflammatory effects through multiple molecular pathways in both in vivo and in vitro studies. BRB’s anti-inflammatory properties make it a promising candidate for further research. Notably, in treating IBDs, BRB has shown effects comparable to or better than salicylate derivatives. Taking into account these reports and the beneficial effect on intestinal microbiota, BRB appears to offer a pleiotropic protective effect on the inflamed intestines. The intestinal microbiota, through its metabolites, may interfere with the functions of the mucous membrane, and excessive bacterial colonization may lead to the translocation of microbes and the induction of diseases, among others, SIBO. The BRB-related restoration of a favorable ratio of beneficial microbial bacteria in the gut also led to a reduction in bacterial colonization-related inflammation. Through this mechanism, a number of beneficial effects of the alkaloid in various internal diseases have been demonstrated, including NAFLD and AGvHD. The mentioned clinical implications, in our opinion, are an important premise for the continuation of research on BRB as a promising alternative to classic anti-inflammatory drugs in the treatment of inflammation, especially in the colon area.

## 8. Materials and Methods

In this review, we searched for articles in the following databases: Scopus, PubMed, Web of Science, and Google Scholar. In total, 126 articles were cited. The articles were qualified for the review by searching using keywords contained in the title and abstract of the article: “berberine”, “colitis”, “inflammation”, and “gut microbiota”. The years of publication of the qualified articles have been reduced to the years after 2012. Figure 1 presents the molecular structure of alkaloid berberine, Figure 2 shows the simplified scheme of main signal pathways that were interfered with by BRB, Figure 3 shows pathways of inflammatory response associated with colitis development, Figure 4 shows the dual effect of BRB on mucin-secreting cells, and Figure 5 shows the BRB-related gut microbiota modulatory effect. Table 1 presents the impact of BRB on the course of internal diseases.

## Figures and Tables

**Figure 1 ijms-26-12021-f001:**
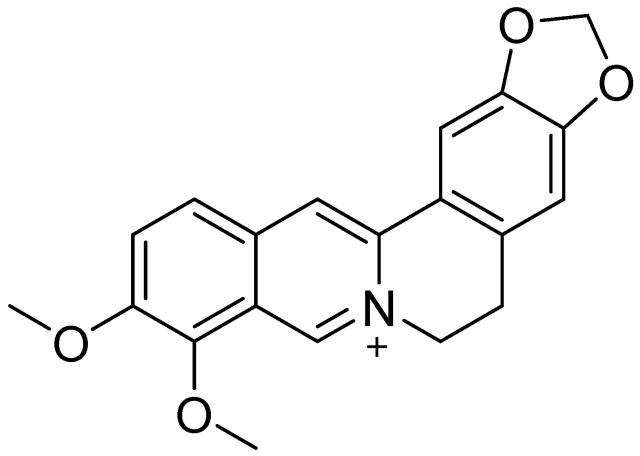
Molecular structure of berberine.

**Figure 2 ijms-26-12021-f002:**
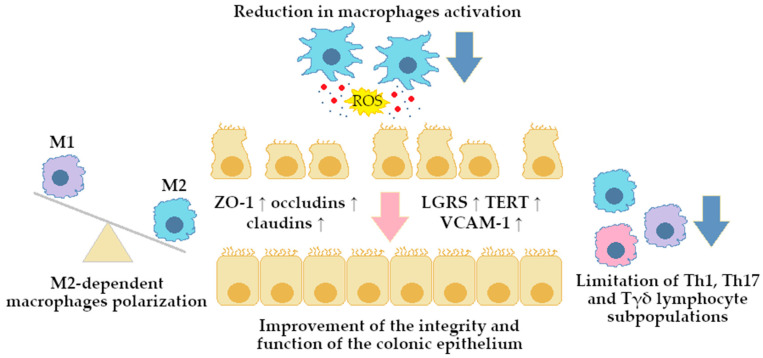
Immunological phenomena in the colon mucosa area following berberine exposure. Abbreviations: LGRS—leucine-rich repeat-containing G protein-coupled receptors, M1/2—macrophages type 1 (pro-inflammatory) and type 2 (anti-inflammatory), ROS—reactive oxygen species, TERT—Telomerase Reverse Transcriptase, Th—T helper cells, Tγδ—T gamma delta cells, VCAM-1—vascular cell adhesion molecule 1, ZO-1—zonula occludens-1. Symbols explanation: blue arrows—down-regulation, pink arrow—process of subsequent changes in mucosa leading to improvement of its integrity, ↑—up-regulation of synthesis/expression.

**Figure 3 ijms-26-12021-f003:**
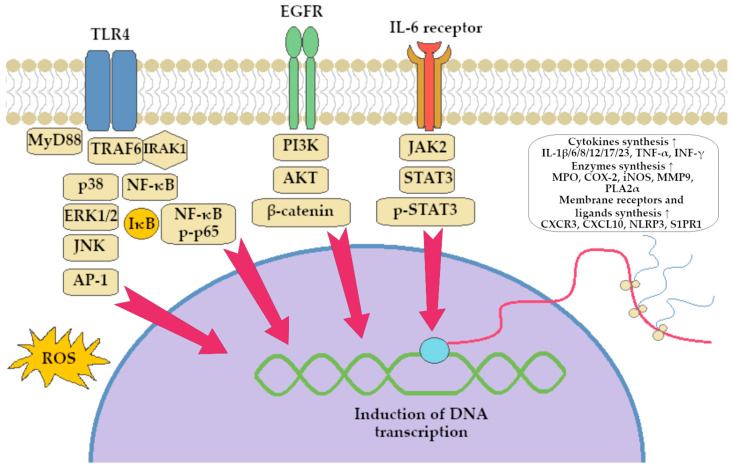
Pathways of inflammatory response associated with colitis development. Abbreviations: AKT—Protein Kinase B, AP-1—Activator Protein 1, COX-2—cyclooxygenase type 2, CXCL10—chemokine (C-X-C motif) ligand, CXCR3—C-X-C chemokine Receptor, EGFR—epidermal growth factor receptor, ERK1/2—extracellular signal-regulated kinases 1/2, IκB—IκB kinase, IL—interleukin, INF—interferone, iNOS—inducible nitrogen oxide synthase, IRAK1—interleukin-1 receptor-associated kinase 1, JAK2—Janus activated kinase 2, JNK—c-Jun N-terminal kinases, MMP9—matrix metallo-proteinase type 9, MPO—myeloperoxidase, MyD88-Myeloid differentiation primary response 88, NF-κB—nuclear factor kappa B, NLRP3—NLR family pyrin domain containing 3, p38—p38 protein, PI3K—phosphoinositide 3-kinase, PLA2α—Phospholipase A_2_ alpha, p-p65—phospho-NF-κB p65, p-STAT3—phosphorylated STAT3 kinase, ROS—reactive oxygen species, S1PR1—sphingosine-1-phosphate receptor 1, STAT3—signal transducer and activator of transcription, TNF—tumor necrosis factor, TLR—toll-like receptor, TRAF6—protein TRAF6. Symbols explanations: red arrows stands for the influence of pointed translation activators on DNA expression, ↑—up-regulation of synthesis/expression.

**Figure 4 ijms-26-12021-f004:**
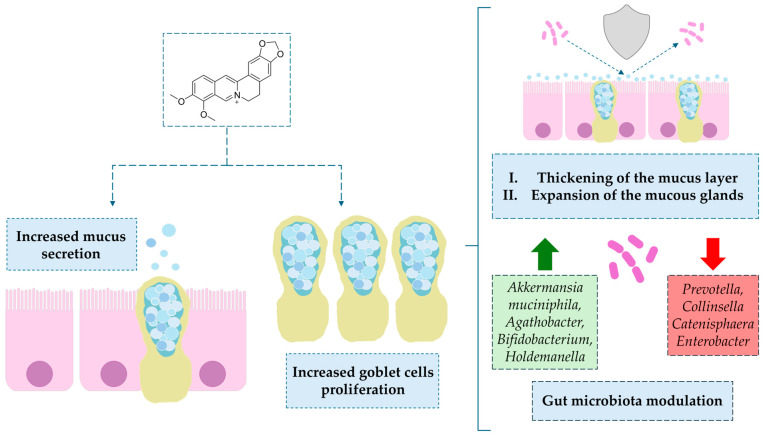
The dual effect of BRB on mucin-secreting cells. Symbols explanations: green arrow—up-regulation of selected bacterial strains, red arrow—down-regulation of selected bacterial strains.

**Figure 5 ijms-26-12021-f005:**
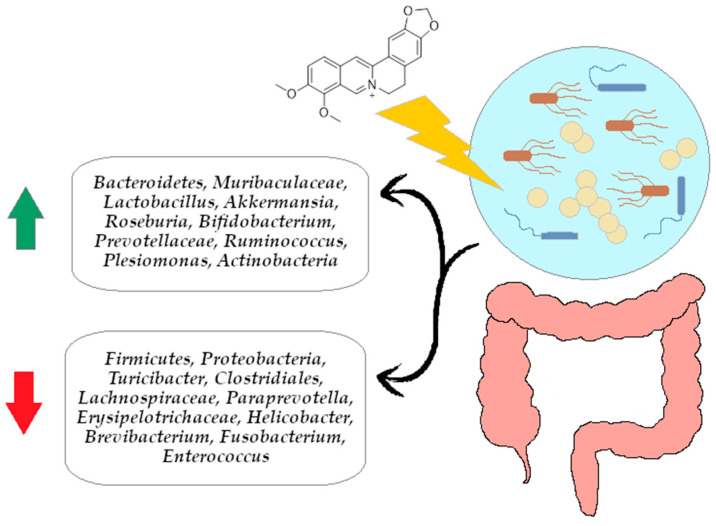
BRB related gut microbiota modulatory effect. Arrows explanations: green arrow—up-regulation of selected bacterial strains growth, red arrow—down-regulation of selected bacterial strains growth.

**Table 1 ijms-26-12021-t001:** BRB impact on several internal diseases with emphasis on its molecular effect.

Disease	Molecular Action of BRB	References
**Intestinal diseases**		
**IBS**	Microbiota modulation: *Proteobacteria*, *Firmicutes ↓* *Bacteroidetes*, *Verrucomicrobiota*, and *Akkermansia ↑*	[101]
	NF-κB signaling pathway silencing	
	VIP, 5-HT signaling ↓	
	AChE activity ↓	
**UC**	Microbiota modulation: *Akkermansiaceae*, *Lactobacillaceae* ↑/*Enterococcus*, and *Ruminococcus* spp. ↓	[71,93,96,97,126]
	Wnt/β-catenin pathway silencing	
	Expression of tight junction proteins: occludins, claudins, ZO-1 ↑	
	Inflammation mediator synthesis: IL-6, IL-17, IL-22, TNF-α, TGF-β ↓/IL-10 ↑	
	Lymphocyte polarization:Th_17_, ILC1 ↓/T_reg_, ILC3 ↑	
	**Non-intestinal diseases**	
**NAFLD**	Microbiota modulation: *Bacteroidetes*, *Proteobacteria **↑**/Firmicutes*, and *Patescibacteria ↓*	[107,108,112,124]
	Apoptosis prevalence ↓	
	Expression of tight junction proteins: occludins, claudins, ZO-1 ↑	
	IL-1β, IL-6, TNF-α, and VCAM-1 mRNA expression ↓	
	MCP-1, CD11c, CD68, and NOS2 mRNA expression ↓	
**PSC**	Bacterial translocation through the intestinal mucosa ↓	[104]
	Amelioration of cholestatic liver injury through modifying the gut–liver axis	
	Restoration of the bile acid homeostasis	
	MCP-1, CD11b, IL-1β, VCAM-1, IL-1α, and CXCL-1 mRNA expression ↓	
**AD**	Microbiota modulation: *Ligilactobacillus*, *Akkermansia*, and *Roseburia* ↑	[116]
	Synthesis of claudins, ZO-1, and occludins ↑	
	ROS generation and neuroinflammation ↓	
**aGvHD**	Microbiota modulation: *Adlercreutzia*, *Dorea*, *Sutterella,* and *Plesiomonas ↑*	[121]
	Alleviation of aGvHD signs and symptoms in the lung, Liver, and colon area	
	IL-1β, IL-6, IL-18, INF-γ, TNF-α, and MCP-1 synthesis ↓	
	NLRP3 inflammasome inhibition	

Legend: AD—Alzheimer’s disease, aGvHD—graft versus host disease, IBS—irritable bowel syndrome, NAFLD—nonalcoholic fatty liver disease, PSC—primary sclerosing cholangitis, UC—ulcerative colitis. Arrows explanations: ↑—up-regulation of synthesis/expression/growth, ↓—down-regulation of synthesis/expression/growth.

## Data Availability

No new data were created or analyzed in this study. Data sharing is not applicable to this article.

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
