# Peer review of "Int. J. Mol. Sci.2025, 26(24), 12021;https://doi.org/10.3390/ijms262412021"

_ijms, 2025, doi:10.3390/ijms262412021_

Round 1

Reviewer 1 Report

Comments and Suggestions for Authors

There is significant repetition between paragraphs starting at line 102 and line 106. Both paragraphs discuss BRB's effects on the nervous system and its potential for IBD and neurodegenerative diseases. These must be consolidated into a single, coherent argument.

Lines 102-117: This entire section is largely redundant with the preceding paragraph (lines 90-99). It should be heavily edited or merged to avoid repetition.

The phrase "Both, *i*n vitro and in vivo studies" has a typo

Typo in "global rinse" should be "global rise"

Typo in "potential as a phototherapeutic adjunct"

The effect of BRB on the Firmicutes/Bacteroidetes ratio is context-dependent (disease model, dosage, duration). The manuscript cannot state it as a universal fact. This section must be rewritten to acknowledge the complexity and context-dependency of these findings, rather than presenting seemingly contradictory results from different studies.

Author Response

Dear Reviewer,

We would like to thank the Reviewer for their time and thorough review of our Manuscript. We have made every effort to respond comprehensively to all comments in order to improve our initial version. We hope that the revised version of the Manuscript will satisfy the Reviewer and meet with his approval and acceptance. Below are our responses to the objections raised.

There is significant repetition between paragraphs starting at line 102 and line 106. Both paragraphs discuss BRB's effects on the nervous system and its potential for IBD and neurodegenerative diseases. These must be consolidated into a single, coherent argument.

Lines 102-117: This entire section is largely redundant with the preceding paragraph (lines 90-99). It should be heavily edited or merged to avoid repetition.

We would like to thank the reviewer for their valuable comment. Lines 102-117 have been rewritten by us. In the corrected version, they read as follows:

„BRB, due to its widespread availability and broad range of activities, seems to be a compound to which science should devote more attention. Of particular interest is the fact that this compound, among other anti-inflammatory activities, also exhibits a number of positive effects on the nervous system. This combination may be significant from the perspective of IBD therapy. Moreover, recent studies suggest that BRB is capable of delaying processes leading to neurodegenerative diseases [34–36]. This promising activity makes BRB a potential candidate for a new anti-inflammatory and immunomodulatory drug used for the treatment of IBD.

This review provides a comprehensive, cross-sectional study combining the molecular mechanisms of BRB's anti-inflammatory action with comparisons of its efficacy to first-line drugs used in IBD (i.e., 5-aminosalicylic acid-5-ASA, sulfasalazine), as well as its effects on the gut microbiota, focusing on the upregulation and downregulation of specific bacterial species. In this way, it fills an information gap regarding the relationship between BRB-related changes and the course of gut-organ axis-dependent diseases, such as nonalcoholic fatty liver disease (NAFLD), ischemic stroke, and graft-versus-host disease (GVHD). Our main goal was to comprehensively present and summarize the current knowledge on the anti-inflammatory properties of this compound in the context of IBD treatment. In addition, we have demonstrated a number of clinical implications that support the continuation of research on this isoquinoline alkaloid as a potential anti-inflammatory and immunomodulatory agent.”

The phrase "Both, *i*n vitro and in vivo studies" has a typo

We would like to thank the reviewer for finding this typo. It has been corrected.

Typo in "global rinse" should be "global rise"

We would like to thank the reviewer for finding this typo. It has been corrected.

Typo in "potential as a phototherapeutic adjunct"

We would like to thank the reviewer for finding this typo. It has been corrected.

The effect of BRB on the Firmicutes/Bacteroidetes ratio is context-dependent (disease model, dosage, duration). The manuscript cannot state it as a universal fact. This section must be rewritten to acknowledge the complexity and context-dependency of these findings, rather than presenting seemingly contradictory results from different studies.

We appreciate this important remark. In order to clarify the ambiguity arising from the description of several studies, we have proposed an additional paragraph (4.1.4) with a brief explanation of why B/F changes varied between studies. Adding an additional section was the most optimal solution for us, as it relates to both sections 4.1.2 and 4.1.3, and we could not do so elsewhere. We encourage the Reviewer to familiarize themselves with the content of the newly added section.

For a more complete picture of the corrections we have made, we encourage the reviewer to familiarize themselves with the latest version of our Manuscript.

Kind regards

Anna Duda-Madej, Przemysław Gagat

Reviewer 2 Report

Comments and Suggestions for Authors

Title: “Berberine in Bowel Health: Anti-inflammatory and Gut-Microbiota Modulatory Effects”. The manuscript presents a well-structured contents of berberine’s anti-inflammatory, microbiota-modulating, and immunoregulatory properties in the context of intestinal and systemic diseases. However, several important scientific and structural issues must be addressed before the manuscript can be considered for further evaluation and publication.

Major comments

  1. The abstract is descriptive but lacks a clear hypothesis or guiding objective for the review. Please clarify.
  2. The current abstract does not address the role of Berberine metabolism by gut microbiota, which is crucial.
  3. Please add brief note of postbiotic relevance where berberine increases SCFAs (important for IBD therapy).
  4. The introduction is detailed but long consider condensing background on general IBD pathophysiology.
  5. Please elaborate content for gut–microbiome-axis connected to berberine.
  6. Author must define the knowledge gap that this review intends to address.
  7. I would recommend mentioning postbiotic therapies (SCFAs, microbial metabolites) https://pubmed.ncbi.nlm.nih.gov/40723826/ because berberine strongly influences SCFA production in IBD.
  8. Clarify whether the review focuses mainly on UC or IBD broadly.
  9. Please provide chemical structural comparison between berberine and its derivatives.
  10. Use a figure or table summarizing microbial taxa increased or decreased by berberine.
  11. More emphasis needed on immune–gut axis-how microbial metabolites influence immune cells https://pubmed.ncbi.nlm.nih.gov/40036891/.
  12. Please include discussion on berberine’s impact on mucin-secreting goblet cells.
  13. Better explain bidirectional gut–immune axis in systemic diseases.
  14. Provide differentiated discussion between acute and chronic inflammation.
  15. Author must discuss limitations of available evidence.
  16. Please highlight research gaps in the Berberine field.

Author Response

Dear Reviewer,

We would like to thank the Reviewer for their time and thorough review of our Manuscript. We have made every effort to respond comprehensively to all comments in order to improve our initial version. We hope that the revised version of the Manuscript will satisfy the Reviewer and meet with his approval and acceptance. Below are our responses to the objections raised.

Major comments

1. The abstract is descriptive but lacks a clear hypothesis or guiding objective for the review. Please clarify.

We thank the Reviewer for this valuable suggestion. In the revised version, we have reformulated the abstract to clearly state the overarching objective of the review. The updated abstract now explicitly highlights that our aim is to synthesize current evidence on:
(i) the anti-inflammatory mechanisms of berberine (BRB) in IBD,
(ii) its comparative efficacy relative to standard first-line therapies, and
(iii) its interactions with gut microbiota, including microbiota-mediated metabolism.
This clarification improves the focus and readability of the abstract, aligning it with the main goals of the review.

2. The current abstract does not address the role of Berberine metabolism by gut microbiota, which is crucial.

We appreciate this important remark. In the revised abstract, we have added a dedicated sentence addressing the metabolic transformation of BRB by gut microbiota into bioactive metabolites. This addition reflects the relevant section presented in the manuscript and acknowledges the role of microbial biotransformation in explaining BRB's high intraluminal activity despite its low systemic bioavailability. The revised abstract now better captures this mechanistic aspect and strengthens its biological relevance.

3. The introduction is detailed but long consider condensing background on general IBD pathophysiology.

We would like to thank the reviewer for pointing this out. We have added supplementary information. It reads as follows:

„The pathogenesis of IBD involves superficial inflammation in UC, originating from these "ulcers," while Crohn's disease progresses throughout the entire length of the substance and is accompanied by the presence of, among other things, granulomas”.

and:

„The underlying process is unclear, but it is likely related to factors that result in an abnormal immune response and, con-sequently, intestinal dysbiosis. Saez 2023”

4. Please elaborate content for gut–microbiome-axis connected to berberine.

We would like to thank the reviewer for pointing out that the content linking berberine to the gut-microbiome axis was not sufficiently emphasized. In response to the reviewer's objection, changes have been made to clarify the effect of BRB on the gut microbiota, with careful consideration given to species that increase and decrease in number. Changes have also been made to the content of Table 1.

In addition to demonstrating the qualitative change in the intestinal flora under the influence of berberine, the effects of this correlation have been systematized, including the impact on the proper synthesis of bacterial metabolites (e.g., short-chain fatty acids, bile acid derivatives, indoles, tryptophan degradation products), which has a significant impact on the prevention of intestinal diseases and the alleviation of their course.

A summary of the information contained in section 4.1.1 has been added.

We invite the reviewer to familiarize themselves with the changes we have made

5. Author must define the knowledge gap that this review intends to address.

Thank you for finding the weak point. We have added this information and refined the wording of our Manuscript's goal, increasing its importance and innovation. It now reads as follows:

This review provides a comprehensive, cross-sectional study combining the molecular mechanisms of BRB's anti-inflammatory action and comparisons of its efficacy with first-line drugs in IBD (i.e., 5-aminosalicylic acid-5-ASA, sulfasalazine), as well as its effect on the gut microbiota, focusing on the up- and downregulation of specific bacterial species. In this way, it fills an information gap regarding the relationship between BRB-related changes and the course of gut-organ axis-dependent diseases, i.e., nonalcoholic fatty liver (NAFLD), ischemic stroke, and graft versus host disease G(VHD). Our main goal was to comprehensively present and summarize the current knowledge on the anti-inflammatory properties of this compound in the context of IBD treatment. In addition, we have demonstrated a number of clinical implications that support the continuation of research on this isoquinoline alkaloid as a potential anti-inflammatory and immunomodulatory agent.

6. Please add brief note of postbiotic relevance where berberine increases SCFAs (important for IBD therapy).

7. I would recommend mentioning postbiotic therapies (SCFAs, microbial metabolites) https://pubmed.ncbi.nlm.nih.gov/40723826/ because berberine strongly influences SCFA production in IBD.

Dear reviewer, thank you for your suggestions in terms of postbiotic effects of berberine application. We added paragraph describing this phenomenon and cited the article pointed:

“SCFAs constitute a large group of organic acids (e.g. butyric acid), referred to as postbiotics - metabolites of microbiome origin, useful in IBD therapy[https://doi.org/10.3390/biom15070954]. The mentioned substances exhibit a pleiotropic mechanism of protective action, interfering with numerous pathways of inflammatory pathogenesis, e.g. via supporting mitochondrial respiration, ATP synthesis, and anti-oxidative effect in the area of colonocytes. Moreover, it has been reported that SCFAs can contribute to the limitation of the colonocytes apoptosis, which allows for the maintenance of intestinal barrier integrity and thus reduces intestinal damage in the inflammatory process [https://doi.org/10.3390/biom15070954].”

8. Clarify whether the review focuses mainly on UC or IBD broadly.

9. Please provide chemical structural comparison between berberine and its derivatives.

Thank you very much for pointing out this gap. Since the subject is very extensive, we have decided not to include it in the present Manuscript but to prepare a new one in the near future. It will be devoted exclusively to berberine derivatives. We will foacus on discussing which chemicall modyfications of this compound are most beneficial in the prevention of gastrointestinal diseases.

10. Use a figure or table summarizing microbial taxa increased or decreased by berberine.

We would like to inform the reviewer that we have made corrections to Table 1, taking into account the effect of berberine on both the increase and decrease in the number of individual bacterial groups in the course of the diseases we discuss, which are directly or indirectly related to the intestines. We invite the reviewer to familiarize themselves with the corrected table.

In addition, we have added a figure illustrating the dual action of berberine on mucin-secreting cells. It is presented as follows:

11. More emphasis needed on immune–gut axis-how microbial metabolites influence immune cells https://pubmed.ncbi.nlm.nih.gov/40036891/.

We would like to thank the reviewer for pointing out this oversight on our part. At the beginning of section 4.2, information has been added on changes in the gut microbiota with age and the impact of this phenomenon on changes in the functioning of the immune system and the progression of neurodegenerative diseases. The detailed mechanism of BRB's impact on Alzheimer's disease is described later in the text, where the influence of bacterial metabolites (mainly SCFAs) on the inhibition of pro-inflammatory pathways and the maintenance of intestinal barrier integrity is discussed. This passage reads as follows:

"SCFAs constitute a large group of organic acids (e.g., butyric acid), referred to as postbiotics—metabolites of microbiome origin, useful in IBD therapy [https://doi.org/10.3390/biom15070954]. The mentioned substances exhibit a pleiotropic mechanism of protective action, interfering with numerous pathways of inflammatory pathogenesis, e.g. via supporting mitochondrial respiration, ATP synthesis, and anti-oxidative effect in the area of colonocytes. Moreover, it has been reported that SCFAs can contribute to the limitation of colonocyte apoptosis, which allows for the maintenance of intestinal barrier integrity and thus reduces intestinal damage in the inflammatory process [https://doi.org/10.3390/biom15070954]."

Similar information on the impact of bacterial metabolites on immune pathways was included in the description of PSC in section 4.1.3. The fragment we added reads as follows:

"At the same time, through a change in Firmicutes, it reduced the activity of bile salt hydrolase (BSH), which leads to a reduction in harmful secondary bile acids, such as deoxycholic acid (DCA) and lithocholic acid (LCA) [95]. BRB also reduced inflammation by downregulating MCP-1, CD11b, IL-1β, VCAM-1, IL-1α, and CXCL1, and inhibiting NF-κB and MAPK signaling pathways  "

12. Please include discussion on berberine’s impact on mucin-secreting goblet cells.

We would like to thank the reviewer for pointing out this serious omission. We have added an entire subsection addressing this issue. It reads as follows:

„BRB exerts beneficial effects on the intestinal environment, in part through its action on mucin-secreting cells. This occurs via two complementary mechanisms: i) regulation of mucin levels, which enhances its protective, mucolytic and/or anti-inflammatory functions [Sikder 2011; Samsuzzaman 2019], and ii) an increase in the number of goblet cells driven by modulation of their proliferation [Dong 2021]. From the perspective of the present study, the second mechanism is of particular relevance. Through this pathway, BRB leads to: i) an increased number of goblet cells and thickening of the mucus layer in the small intestine [Du 2025]; ii) expansion of mucous glands in the colon [Chen 2022], and iii) modulation of the gut microbiota, including an increase in Akkermansia muciniphila, and SCFA-producing bacteria such as Agathobacter, Bifidobacterium, and Holdemanella, along with a reduction in pathogenic taxa such as Prevotella, Collinsella, Catenisphaera, and Enterobacter [Dong 2021; Du 2025].”

13. Better explain bidirectional gut–immune axis in systemic diseases.

Thank you for bringing this point to our attention. Referring to previous points related to this issue, the revised version of our article provides more detailed information on the gut-immune system relationship in the context of systemic diseases (e.g., IBD, IBS – paragraph 2; obesity, insulin resistance, type II diabetes – paragraph 4.1.2; PSC, NAFLD - paragraph 4.1.3; neurodegenerative diseases - paragraph 4.2). A more in-depth interpretation has been provided of the relationship between the qualitative composition of the gut flora, the quality of the metabolites it produces (e.g., SCFA, short-chain fatty acids, bile acid derivatives, indoles, tryptophan degradation products) and the immune system; the impact on both cells and pro- and anti-inflammatory pathways. The impact of the above relationships on the onset and progression of the above-mentioned diseases has also been described.

In addition, a summary of the information collected so far has been added at the end of the article, which is extended by Table 1. We invite the reviewer to familiarize themselves with the latest version of our review. We sincerely hope that the corrections we have made will satisfy the reviewer.

14. Provide differentiated discussion between acute and chronic inflammation.

We would like to thank the reviewer for pointing out this omission. We had trouble deciding where to add it, so we decided to use it as an introduction to Chapter 2. The passage reads as follows:

“Inflammation of the intestines can be acute or chronic. Differentiating is related to the time factor. In acute inflammation, it occurs suddenly, usually within days, and resolves more quickly than in chronic inflammation . Elgazzar2014 Etiological factors are most often viral/bacterial/parasitic, in contrast to the factors of chronic inflammation, which are dominated by autoimmune diseases, drugs, and parasites . Graves2013 The pathogenesis of acute inflammation is based on the participation of neutrophilia, cytokines, macrophages, and T and B lymphocytes. Graves2013 In contrast, chronic inflammation involves T lymphocytes, B macrophages, dendritic cells, and neutophils. Large amounts of cytokines are also produced: IL-23, Th17 TNF, IL-1 β, IFN-γ. Guan2019 They can result in the formation of strictures and even fistulas and abscesses, as in Crohn's disease. Dollinger2024”

15. Author must discuss limitations of available evidence.

Thank you for bringing this to our attention. We have added a chapter addressing this issue at the end of the Manuscript. It reads as follows:

„Despite the large number of literature references cited in this review, we were unable to overlook the limitations related to methodology and interpretation. The use of different DSS/TNBS protocols with varying disease severity makes it difficult to compare results between studies from different research centers and may contribute to differences in the conclusions drawn. In addition, small study samples, short treatment times, and the lack of in-depth microbiome analyses do not allow for unequivocal confirmation of the effectiveness of BRB. It is also worth noting that some studies focused on determining the levels of cytokines, oxidative stress markers, or intestinal barrier proteins without correlating them with the clinical picture, i.e., remission of inflammation or reduction in symptom severity. Furthermore, the Firmicutes/Bacteroidetes ratio remains controversial as it is highly dependent on dietary differences, inflammation levels, intestinal barrier integrity, and metabolic differences. Therefore, studying these correlations without focusing on specific mechanisms is highly inconsistent. These numerous limitations of the data available in the literature further emphasize the need to expand and complexify research on the impact of BRB on the intestinal environment”

16. Please highlight research gaps in the Berberine field.

Thank you for bringing this to our attention. We have added a chapter addressing this issue at the end of the Manuscript. It reads as follows:

„Although it is known that BRB strengthens tight junctions, improves mucosal integrity, and influences goblet cell differentiation, there are still many mechanisms whose molecular basis has not been explained. However, the main obstacle to research on this compound is undoubtedly its low bioavailability and the lack of safety assessment of advanced delivery systems. In the case of the latter, there is an information gap regarding their interaction with the immune system and microbiota. Comprehensive scientific research combining metagenomics, metabolomics, transcriptomics, and proteomics therefore seems necessary. This multi-omic approach will undoubtedly help to understand the action of BRB and enable the determination of the therapeutic window and optimization of the dose of this alkaloid”.

For a more complete picture of the corrections we have made, we encourage the reviewer to familiarize themselves with the latest version of our Manuscript.

Kind regards

Anna Duda-Madej, Przemysław Gagat

Round 2

Reviewer 1 Report

Comments and Suggestions for Authors

Please add researhc gap and novelty in abstarct

Author Response

Dear Reviewer,

We would like to thank you very much for your insightful comment regarding our Abstract. We also did not like it in the version we sent in the revised Article, but we did not have enough time to devote a little more time to it. Therefore, we rewrote it from scratch, and in its current form it reads as follows: 

"Disruption of the gut-microbiome-brain axis contributes to the development of chronic inflammation, impaired intestinal barrier integrity, and progressive tissue damage, ultimately reducing quality of life and increasing risk of comorbidities, including neurodegenerative diseases. Current therapies are often limited by adverse effects and insufficient long-term efficacy, highlighting the need for more comprehensive therapeutic approaches. Berberine (BRB), a plant-derived isoquinoline alkaloid, has attracted growing attention due to its pleiotropic immunomodulatory, neuroprotective, and gut-homeostasis-modulating properties, which involve reshaping the gut microbiota and underscore its therapeutic relevance within the gut-microbiome-brain axis. The aim of this review is to synthesize current scientific evidence regarding the anti-inflammatory mechanisms of BRB in inflammatory bowel disease (IBD). We compare its activity with first-line therapies and discuss its impact on microbial composition, including the bidirectional regulation of specific bacterial taxa relevant to intestinal and systemic disorders that originate in the gut. Furthermore, we emphasize that gut bacteria convert BRB into bioactive metabolites, contributing to its enhanced intraluminal activity despite its low systemic bioavailability. By integrating molecular and microbiological evidence, this review fills a critical knowledge gap regarding the comprehensive therapeutic potential of BRB as a promising candidate for future IBD interventions. The novelty of this work lies in unifying fragmented findings into a framework that explains how BRB acts simultaneously at the levels of host immunity, microbial ecology, and neuroimmune communication – thus offering a new conceptual model for its role within the gut-microbiome-brain axis."

We hope that in its current form it meets all your expectations.

Kind regards

Anna Duda-Madej

Przemysław Gagat

Reviewer 2 Report

Comments and Suggestions for Authors

I appreciate that the authors have meticulously revised the manuscript, and I would now recommend it for publication.

Author Response

Dear Reviewer,

Thank you very much for your time, which has allowed us to improve our manuscript.

Kind regards

Anna Duda-Madej

Przemysław Gagat